# Automatic Damage Detection of Pavement through DarkNet Analysis of Digital, Infrared, and Multi-Spectral Dynamic Imaging Images

**DOI:** 10.3390/s24020464

**Published:** 2024-01-11

**Authors:** Hyungjoon Seo, Yunfan Shi, Lang Fu

**Affiliations:** 1Department of Civil and Environmental Engineering, University of Liverpool, Liverpool L69 7WW, UK; l.fu8@liverpool.ac.uk; 2Department of Computer Science, University of Liverpool, Liverpool L69 7WW, UK; sgyshi31@liverpool.ac.uk

**Keywords:** automatic damage detection, pavement, DarkNet, IR image, MSX image, wavelet transform

## Abstract

It is important to maintain the safety of road driving by automatically performing a series of processes to automatically measure and repair damage to the road pavement. However, road pavements include not only damages such as longitudinal cracks, transverse cracks, alligator cracks, and potholes, but also various elements such as manholes, road marks, oil marks, shadows, and joints. Therefore, in order to separate categories that exist in various road pavements, in this paper, 13,500 digital, IR, and MSX images were collected and nine categories were automatically classified by DarkNet. The DarkNet classification accuracies of digital images, IR images, and MSX images are 97.4%, 80.1%, and 91.1%, respectively. The MSX image is a enhanced image of the IR image and showed an average of 6% lower accuracy than the digital image but an average of 11% higher accuracy than the IR image. Therefore, MSX images can play a complementary role if DarkNet classification is performed together with digital images. In this paper, a method for detecting the directionality of each crack through a two-dimensional wavelet transform is presented, and this result can contribute to future research on detecting cracks in pavements.

## 1. Introduction

Damages such as cracks and potholes in road pavements can be factors that cause accidents on the road. Even if the damage is small, it can be worsened by rain and have a large impact on vehicles if colliding with running vehicles [1]. Therefore, various studies have been conducted, including manual inspection methods, to detect damage to road pavements [2,3]. However, manual inspection is not only inefficient but also cannot cover all roads. Tertre et al. (2020) conducted a study to predict the master curve of asphalt pavement with ultrasonic as a representative during non-destructive testing [4]. Tran and Roesler (2022) conducted a study to detect concrete joints in road pavements with shear wave transmission energy [5]. Chen et al. (2022) used an accelerometer to automatically detect transverse cracks in road pavements through a WT-CNN [6]. Damage detection of road pavements using various sensors can accurately detect damage, but there is a limit in its application to city or country locations.

Recently, studies on automatically detecting damage on road pavements through deep learning or machine learning of captured images have been actively conducted. In many disciplines, machine learning classifiers that rely on effective feature engineering have been reported to be successful, starting from extracting features such as texture, gray-level co-occurrence matrix (GLCM), or image segments via methods like Otsu and then using a support vector machine (SVM), artificial neural network (ANN), or Random Forest (RF) to classify cracked pavements. Hu et al. (2010) found that the texture features of cracked pavement tend to be uneven. Therefore, an SVM model can be used to classify the image into cracks and noncracks [7]. Cord and Chambon (2012) devised a more general supervised learning approach using the AdaBoost classifier [8]. Shi et al. (2016) designed a random forest for asphalt crack pavement extraction, which is valid also in small, supervised datasets for classification [9]. Hoang and Nguyen (2019) have used a combination of an SVM, ANN, and RF in the training and verification process to measure the performance of the model in the dataset [10]. Chen et al. conducted a study to find crack areas through image processing with multiple thresholds [11]. Research on automatic crack detection is being conducted not only regarding pavements but also in various other fields. In heritage buildings, maintenance is performed based on thermal infrared images and point cloud data [12,13]. In the field of tunnels and pillars, research has not only been conducted to predict joints and cracks in tunnel faces and pillars [14], but research has also been conducted to predict the ground in front of the tunnel through artificial intelligence technology [15]. 

Multi-type crack classification is often performed using deep learning methods like convolutional neural networks (CNNs). In many cases, the process starts with pretrained models such as VGG-16, Inception-v3, Xception, and VGG-19/Resnet152. In addition, DarkNet-53, which belongs to the DarkNet series, is used as the backbone network in the YOLOv3 family, which has been applied to detect pavement cracks with high accuracy [16,17]. Despite the fact that deep learning is more computationally expensive than other methods, an accuracy of above 90% can be achieved after data augmentation and fine tuning. Zou et al. (2018) designed an encoder–decoder style deep convolutional neural network (DCNN) on hierarchical multi-scale input features with a test set F-score of roughly 0.87 [18]. A DCNN fusion model combining the advantages of the U-Net model and the multitarget single shot multibox detector (SSD) CNN model is proposed by Feng, X et al., where classification is performed after segmentation. The detection accuracy of the system is above 85%, though a relatively larger number of parameters are handled [19]. Chen et al. (2022) defined the characteristics of each frame as Local Binary Patterns (LBP) and Principal Component Analysis (PCA) in the filmed video and then classified the damage to the road pavement through a Support Vector Machine (SVM) [20]. Ibragimov (2022) conducted a study on detecting distress in the pavement through convolutional neural networks, and Jiang et al. (2022) detected and segmented cracks in road pavements through a two-step deep learning approach [21,22]. Chandra et al. (2022) studied the seasonal effect to detect complex pavement defects and Chen et al. (2022) used thermal images to detect pavement damages [23,24]. Recently, studies on locating damage such as potholes through 3D data are also being conducted [25]. 

In this paper, not only digital images of road pavements are collected but also IR and MSX images based on thermal infrared imaging. We tried to find damage such as longitudinal and transverse cracks, crocodile cracks, and potholes by classifying images taken with the DarkNet Classifier into each category. It is expected that this study will be able to contribute to research on detecting damage to road pavements through digital-based images, as well as thermal infrared images of road pavements. Thermal infrared images can detect damage that is not able to be found in digital image analysis, and the two data can complement each other for damage detection. Therefore, through the research of this paper, various types of damage and other types of structures existing on road pavements can be automatically detected.

## 2. Data Collection for Pavement Damage Detection

### 2.1. Applied Photogrammetry Sensors 

In this study, the following three images were used for data acquisition: digital images; infrared (IR) images; and Multi-Spectral Dynamic Imaging (MSX) images. To acquire digital image data, a mobile phone was used and an FLIR One Pro LT camera from FLIR (West Malling, UK) was used to collect IR images. This camera has an additional MSX imaging option wherein visible light details are added onto the thermal image for clarity enhancement. FLIR MSX adds visible light details to thermal images in real time for greater clarity, embedding edge and outline details onto thermal readings. Unlike image fusing (the merging of visible light and thermal image), MSX does not dilute the thermal image or decrease thermal transparency. The collected images each have different image characteristics, and Figure 1a shows a digital image. The IR image shows the distribution of thermal energy, and cracks can be found because there is different thermal energy between the crack and the pavement (see Figure 1b). The MSX image is a more enhanced IR image, and it can be seen that the intensity of color between pixels is enhanced compared to the IR image (see Figure 1c). In this paper, by analyzing three different images with the same algorithm, it can be determined which pavement categories were well identified. The specification details of the infrared thermal camera are shown in Table 1.

### 2.2. Field Test

Field tests were conducted in Liverpool, UK, and various elements such as shadows and oil marks existed in the road pavement, as well as damage. These factors can be misclassified for damage by the deep learning algorithm. Therefore, in order to reduce the misclassification of the deep learning algorithm as much as possible, the various factors existing in the road pavement were all included in the category of deep learning. Categorization of deep learning and machine learning generally requires about 100 or more images [26]. In this paper, to ensure high prediction accuracy, 500 images were collected by dividing road pavement elements into nine categories. The categories classified as damage were alligator cracks, longitudinal cracks, transverse cracks, joints, and potholes, as shown in Figure 2a, and the categories classified as non-damage, shown in Figure 2b, were manholes, oil marks, road marks, and shadows. In total, 13,500 images were collected because three different images were collected identically for the nine different categories (see Figure 2).

### 2.3. Color Scale Analysis for Raw Image Data

In this paper, 300 digital, IR, and MSX images each were collected for nine categories of pavement types. Images corresponding to each category have unique features representing the categories according to the distribution of pixel color values in the image. Therefore, the color value distribution of pixels in each image is represented as a histogram. Figure 3 shows the distribution of pixel values when analyzing the digital image of the manhole. In the digital image of Figure 3a, it can be seen that gray is mainly concentrated, rather than red and blue colors. The histogram of the red scale clearly differentiates the gray values representing manholes and pavements. In the histogram of the red color scale, 255 represents pure red and 0 represents black color. For a manhole that is dark gray, most of the pixels are distributed between 105 and 165 on the red color scale. However, pixels representing the road pavement exist on the red color scale between 60 and 105 (see Figure 3b). In the distribution of the red color scale according to the location of the pixel, it can be confirmed in three dimensions that the red color scale is highly distributed in the manhole (see Figure 3c). Figure 3d shows the histogram of the blue color scale. In the histogram of the red color scale, 255 represents pure blue and 0 represents black color. Pixels representing manholes are distributed between blue scale values of 102 and 160, and pixels representing road pavements are distributed between blue scale values of 60 and 102. When the blue scale value is displayed at each pixel position, the manhole is not clearly distinguished (see Figure 3e).

Figure 4 shows an example of a histogram analysis for IR images of manholes. Since the temperature difference between the manhole and the road is different, Figure 4a shows the color distribution between the two materials. Figure 4b shows the histogram of the red scale, where 255 represents pure red and 0 represents black color. Pixels representing manholes are intensively distributed between red scales 0 and 25, and pixels representing pavements are widely distributed between 115 and 255. As shown in Figure 4c, when the red scale distribution according to the pixel position is viewed in three dimensions, the position and shape of the manhole are more clearly revealed. Pixels representing the pavement have a high red scale, and pixels representing manholes have a red scale close to zero. Figure 4d shows the histogram of the blue scale, which has an opposite result to the red scale histogram. Pixels representing road pavements are distributed between 0 and 55 on the blue scale, and pixels representing manholes are distributed between 140 and 255. The three-dimensional histogram of the blue scale also shows the opposite result to that of the red scale (see Figure 4e). It shows that pixels representing the manhole tend to reach values close to 255 on the blue scale, and pixels representing the pavement have values close to 0.

Figure 5 shows an example of the histogram analysis for MSX images of manholes. MSX images are pixel-enhanced images of IR images because MSX is an imaging technology that produces detailed thermal images by bringing together the visual and thermal spectrums. Figure 5a shows the MSX image of the manhole, which shows clearer boundaries between materials with different thermal energies than the IR image. Figure 5b shows the histogram results for the red scale. As shown in the histogram results, pixels representing the pavement are intensively distributed on the red scale at a value of 255, and pixels representing the manhole are intensively distributed on the red scale at a value of 0. Unlike the histogram result of the IR image, it can be seen that the boundary between the pixels representing the pavement and the manhole is clearly divided due to the enhancement of the color of the image. In the result of the three-dimensional histogram, the red scale value reached 255 at the location of pixels representing the pavement, and the values representing the manhole reached 0 (see Figure 5c). In particular, in the IR image analysis result, there are less than 1000 pixels corresponding to pure red, but in the MSX analysis result, more than 24,000 pixels showed values corresponding to pure red. The histogram result of the blue scale appeared opposite to the result of the red scale, and the difference between the blue scale values of the pixels representing the pavement and the manhole appeared more clearly than the IR image result (see Figure 5d). In particular, more than 65,000 pixels representing the pavement are concentrated in ‘0’. The distribution of pixels representing the manhole and the pavement is clearly separated at both ends of the histogram. Figure 5e shows the three-dimensional histogram result of the blue scale, and the location of the manhole and the location of the pavement are clearly revealed. Pixels representing manholes soar in the three-dimensional histogram as the blue scale value is close to 255, and pixels representing the pavement are located at the bottom of the histogram as the blue scale value is close to 0. The change patterns of these two or three-dimensional histograms act as an important feature enabling categorization in neural network frameworks such as DarkNet.

## 3. Neural Network for Automatic Damage Detection

DarkNet-19 is a DarkNet structure with 19 basic units, each of which consists of a three-dimensional convolution layer plus a batch normalization layer and a leaky relu activation layer, with different parameters such as filter/output size and stride [27,28] (see Figure 6). Convolution layers are responsible for feature extraction and dimensionality reduction (as depicted in the graph), while batch normalization scales learnable weights and biases of each neuron into (0, 1, inclusive), which makes training faster and more stable in terms of gradient update. Finally, leaky relu is a preferred choice of activation function for the result and behaves more stably in the training process of deep neural networks with a large number of training iterations compared to classic relu or sigmoid. Until this layer, all weights and biases start from pre-trained values. After the last (19th) single convolution layer, Global Average Pooling is used to replace classic fully connected layers to generate class prediction confidence. Finally, based on the confidence, a softmax layer generates final class predictions. Notice that the input/output layer dimension needs to be updated based on data type (on summer days, nine classes, otherwise eight).

## 4. Automatic Damage Detection

### 4.1. DarkNet Result

Figure 6 shows the results of DarkNet’s confusion matrix for nine categories of pavement damage. The digital image showed an average accuracy of 97.4% for the nine categories (see Figure 7a). The detection result of longitudinal cracks was the lowest at 89.3%, and manholes, oil marks, potholes, and shadows showed 100% accuracy. Figure 7b shows that the average accuracy of the nine categories is 80.1% as a result of DarkNet’s confusion matrix for the IR camera. The transverse crack showed the lowest accuracy at 74.0%, and the highest accuracy was 81.3% with the shadow. The results of the MSX images showed that the average accuracy of the nine categories was 91.1% (see Figure 7c). The maximum accuracy was 100% with the manhole, and the minimum accuracy was 80% with the longitudinal crack.

Figure 8 shows a polygon chart for a detailed comparison of the results of three images. It can be seen that the digital image has the highest accuracy in all categories, and the manhole has the highest accuracy in all images. The tendency of accuracy in each category was found to decrease and increase similarly. Among damages, potholes showed relatively high accuracy except for the IR image results, but longitudinal and transverse cracks showed low accuracy in the results of all images, compared to the other categories. In particular, the classification accuracy of transverse cracks in IR images was the lowest. The MSX images had higher accuracy in all categories than the IR images, with about 11% higher accuracy on average. The MSX image had a slightly lower accuracy than the digital image result, and it showed about 6% lower accuracy on average. Therefore, if supplementary analysis is required for DarkNet analysis of digital images, analysis through MSX images can ensure high accuracy. MSX images can therefore be used as a dataset to complement digital images in automatic damage detection systems.

### 4.2. Analysis of Misclassified Images

DarkNet analysis results of digital images had the highest accuracy but made classification errors in certain categories, and Figure 9 shows examples of the misclassified digital images in the DarkNet analysis. Figure 9a is a digital image of a longitudinal crack, but it is misclassified as a joint. The image in Figure 9b was taken of a transverse crack but was misclassified as an oil mark. The surface of the pavement is generally rough, and especially when water exists on gravel and uneven surfaces, pixels in this part present low-brightness images. Therefore, water present on the pavement is a factor that causes classification errors in digital images, and taking digital images on a sunny day can increase the classification accuracy. In Figure 9c, potholes and both longitudinal and transverse cracks exist, but DarkNet classifies them as longitudinal cracks. In Figure 9d, there are shadows and small transverse cracks, but DarkNet classifies them as transverse cracks. DarkNet is not able to select multiple categories when there are multiple categories in one scene. Therefore, it is difficult to expect high accuracy of the DarkNet in a section of pavement where multiple categories exist.

The IR image shows the lowest accuracy among the three images, and Figure 10 shows examples of misclassified IR images. Figure 10a is an IR image of the joint, but it is misclassified as a pothole. The temperature change between the joint and the pavement is clearly visible in the IR image and the boundary of the joint is clearly distinguished. However, as the temperature inside the joint appears lower than that of the pavement, a pothole-like shape is detected and classified as a pothole. Figure 10b is an IR image of a longitudinal crack, but it is misclassified as a shadow. In the IR image, the temperature change of the longitudinal crack is lower than that of the pavement. However, it is misclassified as a shadow because the thickness of the longitudinal crack is expressed as thick. Figure 10c is an IR image of an oil mark, but it is misclassified as a pothole. IR images show an accuracy of 78% and 76%, with oil marks and potholes, respectively, and most of the examples are misclassified between the two categories. Therefore, it is difficult to apply IR images to distinguish between shadows and potholes. Road marks also have a relatively low accuracy, with 78% classification accuracy in IR images, and are mostly classified as shadows (see Figure 10d). In the digital image, the road marks are distinguished by showing a low color scale, such as white. However, about 22% of the IR images are misclassified because they show almost the same temperature change and shape as the shadow in the IR image. It is found that the resolution of the IR image is also one of the factors that has an important influence on the classification accuracy. In the digital image, the alligator cracks are clearly distinguished, but the low resolution of the IR image is not able to capture the distribution pattern of the alligator cracks in the image in detail. Therefore, the temperature changes of the microcracks in the alligator cracks are integrated as a shape and misclassified as shadows (see Figure 10e). As with digital image results, if an image has two different elements, DarkNet will choose one classification. Figure 10f includes both a pothole and transverse crack, but DarkNet classified this IR image as a transverse crack.

MSX images have a higher accuracy than IR images and lower accuracy than digital images, and examples of misclassified MSX images are shown in Figure 11. Infrared thermal imaging is greatly affected when there is water seeping into the pavement. Figure 11a is an MSX image of the pothole, but it is misclassified as a transverse crack due to the water effect around the pothole. In Figure 11b, the area maintaining low temperature is widely distributed because water permeates the pavement around the transverse crack. Therefore, this MSX image was misclassified as a longitudinal crack due to the scattered water. Figure 11c is an MSX image of the shadow, but since the thickness of the shadow is small, it is misclassified as a transverse crack. If the difference in thickness and temperature between the crack and the shadow is not large, the shadow can be misclassified as a transverse crack. Figure 11d is a road mark but is misclassified as a transverse crack. In the digital image, the road mark has a low color scale like white, and is distinguished from the pavement, but in the thermal infrared image, the temperature of the road mark is lower than that of the road pavement, so it is classified as a crack. Figure 11e shows alligator cracks that are distributed in the longitudinal direction but are misclassified as longitudinal cracks. The MSX images showed about 16% higher classification accuracy of alligator cracks than the IR images, detecting micro-cracks better than IR images. However, it is difficult to classify alligator cracks when micro-cracks are combined into a single shape of the image, as shown in Figure 11e. Figure 11f is an oil mark but is misclassified as a pothole. Potholes and oil marks are not only similar in shape, but in this case, the temperature is lower than that of the pavement.

### 4.3. Wavelet Analysis

Wavelet transform is a mathematical tool used in signal processing and image analysis to analyze and represent signals or images in terms of different frequency components. The two-dimensional wavelet transform is particularly useful for image processing applications. Detecting the directionality of cracks in images is an important task in structural health monitoring and non-destructive testing. The first step is to represent the image containing cracks using a two-dimensional wavelet transform. This involves decomposing the image into different frequency components and extracting information about the direction and magnitude of features within the image. The image is transformed using a 2D wavelet transform, which involves convolving the image with wavelet functions at different scales and orientations. The avelet transform decomposes the image into approximation coefficients (low-frequency information) and detail coefficients (high-frequency information) in both the horizontal and vertical directions. The detail coefficients obtained from the wavelet transform contain information about the high-frequency components of the image in different directions. Cracks often introduce high-frequency patterns in images, and the orientation of these patterns can provide clues about the direction of the cracks. By analyzing the magnitude and orientation of the detail coefficients, one can identify the dominant orientations in the image. The dominant orientation of the high-frequency components can be used to detect the directionality of cracks in the image. For example, if there is a significant presence of high-frequency details in a particular direction, it indicates that there might be a crack aligned with that direction. Thresholding techniques can be applied to filter out noise and retain only the significant features related to cracks. By setting appropriate thresholds on the wavelet coefficients, one can enhance the visibility of cracks and suppress irrelevant details. The results can be visualized to highlight the detected cracks and their orientations. Further analysis, such as connecting adjacent pixels with significant orientations, can be performed to identify the complete structure of cracks in the image. In summary, the two-dimensional wavelet transform provides a multi-resolution analysis of image features, making it a powerful tool for detecting the directionality of cracks in images. The orientation information obtained from the wavelet transform helps in identifying the orientation of cracks and enhances the overall crack detection process. The equations of wavelet transform formulas used in this paper are as follows:(1)$$W_{j,k}^{H}\left(u,\;v\right)=\sum_{m=0}^{1}\sum_{n=0}^{1}h_{m,n}\cdot W_{j+1,2k+m}^{H}\left(u,\;2v+n\right)$$
(2)$$W_{j,k}^{V}\left(u,\;v\right)=\sum_{m=0}^{1}\sum_{n=0}^{1}h_{m,n}\cdot W_{j+1,2k+n}^{V}\left(2u+m,\;v\right)$$
(3)$$W_{j,k}^{D}\left(u,\;v\right)=\sum_{m=0}^{1}\sum_{n=0}^{1}h_{m,n}\cdot W_{j+1,2k+m}^{D}\left(2u+m,\;2v+n\right)$$
(4)$$W_{j,k}^{A}\left(u,\;v\right)=\sum_{m=0}^{1}\sum_{n=0}^{1}h_{m,n}\cdot W_{j+1,2k+n}^{A}\left(2u+m,\;2v+n\right)$$
where $W_{j,k}^{H}\left(u,\;v\right)$, $W_{j,k}^{V}\left(u,\;v\right)$, $W_{j,k}^{D}\left(u,\;v\right)$, and $W_{j,k}^{A}\left(u,\;v\right)$ are the horizontal, vertical, diagonal, and approximation coefficients at the scale j and position k in the two-dimensional wavelet transform, respectively. $h_{m,n}$ represents the filter coefficients of the wavelet filter. u and v are the spatial coordinates in the transformed domain. The indices j and k represent the scale and position, respectively. This equation describes the recursive relationship between the wavelet coefficients at different scales and positions in terms of the filter coefficients $h_{m,n}$ and the wavelet coefficients at a higher scale. The wavelet transform is applied iteratively to obtain coefficients at multiple scales, providing a multi-resolution representation of the input signal or image.

This paper proposes ways to minimize image misclassification of DarkNet through detailed image analysis. It is not able to find the cause of misclassification because DarkNet recalculates the image for each layer and makes its own classification decision by training. Figure 12a shows a collage of all image slices of cubes representing activation outputs per layer in the IR image of the oil mark. The output of layer 7 is shown in Figure 12b, and this could cause it to be misclassified as a pothole. However, in this process, it is not possible to infer the clear reason for misclassification or the decision of classification in each layer.

In this paper, an in-depth analysis was conducted to prepare a plan to reduce misclassification, and the results of the wavelet transform analysis are expected to be helpful in finding the direction of cracks in road pavements in the future. Wavelet transform is an effective tool for time-frequency analysis. The input image is subjected to a two-dimensional discrete wavelet transform (DWT). The image is decomposed into approximate, horizontal, vertical, and detailed coefficient matrices. The approximate coefficients represent low-frequency components, whereas the detailed coefficients (horizontal, vertical, and diagonal) represent the high-frequency components, which provide both high-frequency values for both the background and foreground, like edge and non-edge regions [29]. Figure 13 shows an example of the DWT of an IR image in the joint. The horizontal coefficient can physically filter out the horizontal components of a joint. Therefore, as shown in Figure 13, the horizontal coefficient can extract the horizontal boundary between the joint and the pavement. The vertical coefficient physically detects the vertical components of a joint. The pixels of the vertical components representing the joint in the IR image are detected by the vertical coefficient. The diagonal coefficient detects the diagonal components of the image, and the pixels representing the diagonal components of the joint are shown in Figure 13.

In the DarkNet results of the three images, the transverse cracks and longitudinal cracks corresponding to the damage of the pavement had lower accuracy than the classification accuracy of other categories. Detecting transverse, longitudinal, and alligator cracks in pavements is the ultimate purpose of this study and clearly identifying the directionality of cracks not only increases the classification accuracy but also contributes to future automatic crack detection studies using deep learning. Therefore, in this paper, an analysis was conducted to detect the directionality of the crack by wavelet transforming the alligator crack, which has horizontal, vertical, and diagonal components (see Figure 14). Alligator cracks in the digital image are clearly detected by the horizontal and vertical coefficients. In Figure 14a, horizontal components of alligator cracks are distributed in the upper part of the figure, and vertical components of alligator cracks are distributed in the lower part. Due to the nature of alligator cracks, there are inevitably many diagonal components, which are also clearly detected. In the IR image, the vertical component is clearly revealed, and the horizontal and diagonal components are also found in several places in the image, revealing the features of alligator cracks (see Figure 14b). In MSX images, the vertical components are observed more closely than in IR images. Since alligator cracks are spread throughout the image, the horizontal and diagonal components are also found widely in the image (see Figure 14c). Cracks are a typical damage feature of pavements, and the DWT results on the directionality of these cracks will increase the accuracy of classifying the crack itself, as well as the accuracy of classifying the type of cracks in future studies.

Predicting the direction of cracks is expected to contribute to future automatic crack detection research. In particular, predicting the directionality of cracks can have a significant impact on distinguishing between cracks and other structures such as joints. Additionally, because the propagation of cracks can be predicted, providing direction can also contribute to future road pavement maintenance.

## 5. Conclusions

In this paper, a study was conducted to automatically detect road pavement damage by analyzing digital, IR, and MSX images of the pavement with DarkNet. The detailed conclusions of this study are as follows.

(1)In total, 13,500 digital, IR, and MSX images were collected for the following nine categories of the pavement: longitudinal cracks, transverse cracks, alligator cracks, potholes, shadows, joints, road marks, manholes, and oil marks. Nine categories were automatically classified with the DarkNet-19 classifier.(2)The digital images showed an average accuracy of 97.4% for the nine categories. The detection result of longitudinal cracks was the lowest at 89.3%, and manholes, oil marks, potholes, and shadows showed 100% accuracy. The average accuracy of the nine categories was 80.1% as a result of DarkNet’s confusion matrix of the IR camera. Transverse cracks showed the lowest accuracy at 74.0%, and the highest accuracy was 81.3% with shadows. The results of the MSX images showed that the average accuracy of the nine categories was 91.1%. The maximum accuracy was 100% with manholes, and the minimum accuracy was 80% with longitudinal cracks.(3)The classification accuracy of transverse cracks in the IR images was the lowest. The MSX images had a higher accuracy in all categories than the IR images, with about 11% higher accuracy on average. The MSX images had a slightly lower accuracy than the digital images, and they showed about 6% lower accuracy on average. Therefore, if supplementary analysis is required for DarkNet analysis of digital images, analysis through MSX images can ensure high accuracy.(4)An in-depth analysis was conducted on misclassified digital, IR, and MSX images. Water on the pavement had the greatest effect on the misclassification of the three images. DarkNet inevitably causes misclassification when multiple categories exist in one scene. In the IR and MSX images, misclassification occurred due to the influence of the resolution of the thermal infrared image.

Detecting transverse, longitudinal, and alligator cracks in the pavement is the ultimate purpose of this study, and clearly identifying the directionality of cracks not only increases the classification accuracy but also contributes to future automatic crack detection studies using deep learning. Therefore, in this paper, an analysis was conducted to detect the directionality of the crack by wavelet transforming. Cracks are typical forms of pavement damage, and the DWT results on the directionality of cracks will increase the accuracy of classifying cracks itself, as well as the accuracy of classifying the type of cracks in future studies. Future research will be to develop an algorithm that can improve the accuracy of automatic damage detection by complementing each other with three different images.

## Figures and Tables

**Figure 1 sensors-24-00464-f001:**
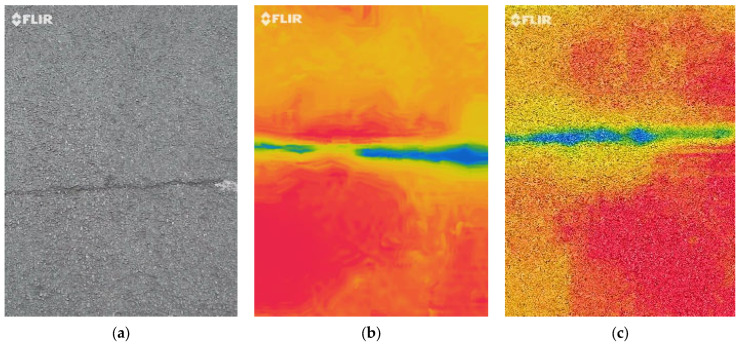
Examples of three different image data. (**a**) Digital image, (**b**) IR image, (**c**) MSX image.

**Figure 2 sensors-24-00464-f002:**
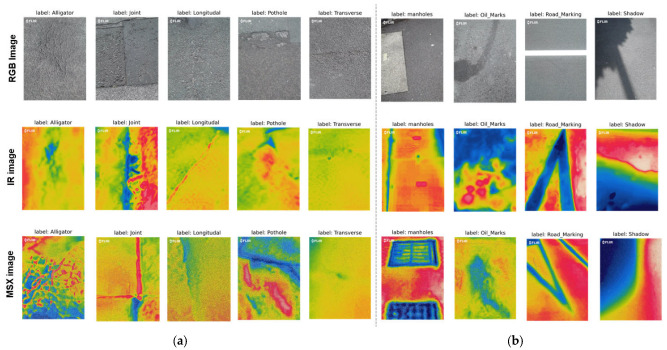
Raw image data. (**a**) Categories representing damages. (**b**) Categories representing non-damages.

**Figure 3 sensors-24-00464-f003:**
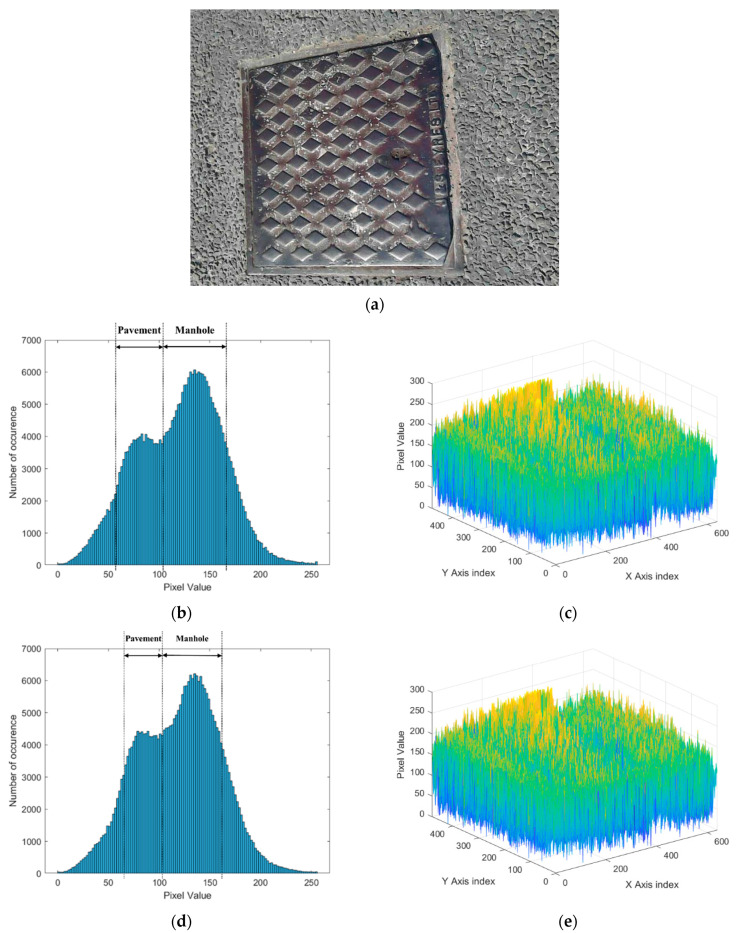
Example of digital image analysis of manhole. (**a**) Digital image. (**b**) Histogram of red color. (**c**) Three-dimensional histogram of red color in image. (**d**) Histogram of blue color. (**e**) Three-dimensional histogram of blue color in image.

**Figure 4 sensors-24-00464-f004:**
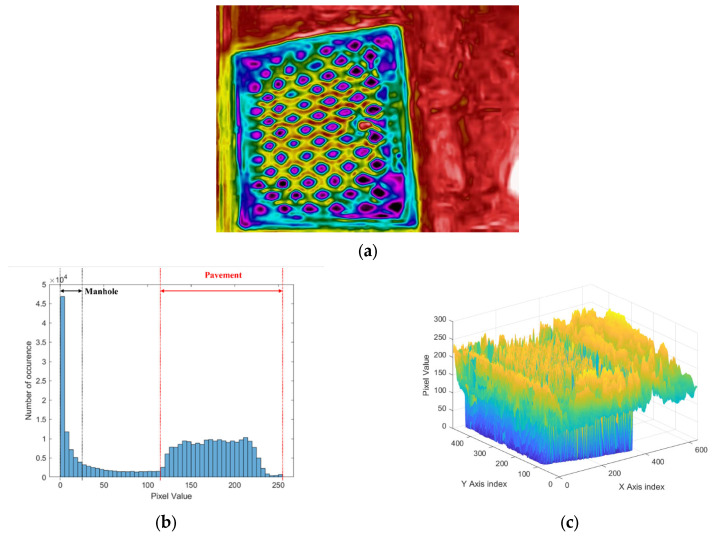
Example of IR image analysis in manhole. (**a**) IR image. (**b**) Histogram of red color. (**c**) Three-dimensional histogram of red color in image. (**d**) Histogram of blue color. (**e**) Three-dimensional histogram of blue color in image.

**Figure 5 sensors-24-00464-f005:**
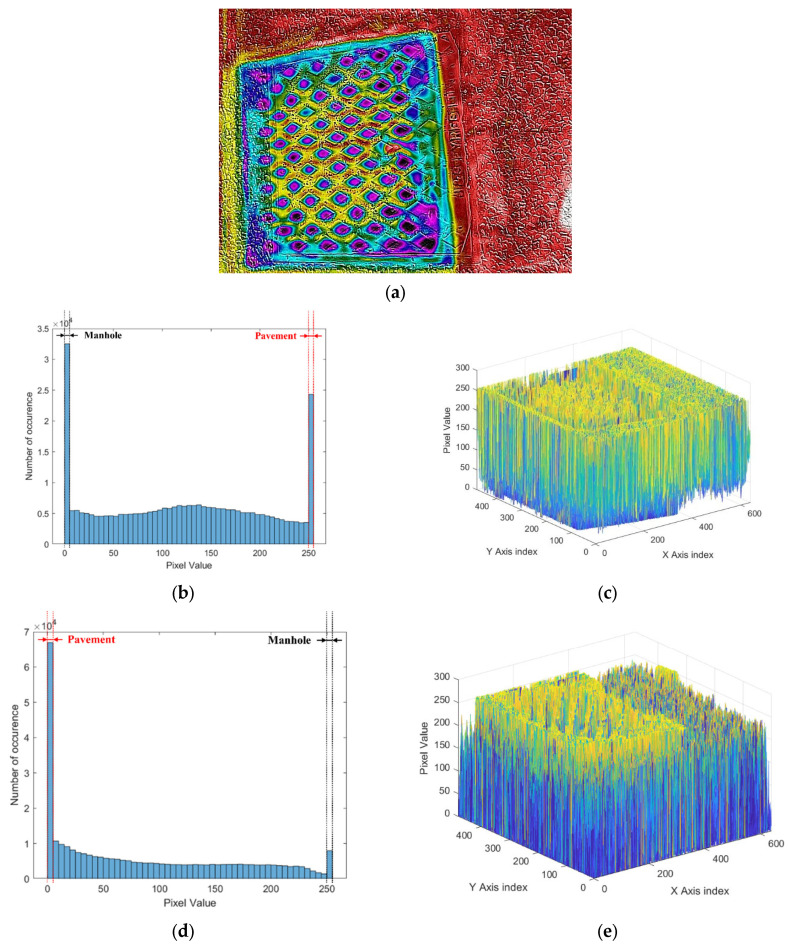
Example of MSX image analysis in manhole. (**a**) MSX image. (**b**) Histogram of red color. (**c**) Three-dimensional histogram of red color in image. (**d**) Histogram of blue color. (**e**) Three-dimensional histogram of blue color in image.

**Figure 6 sensors-24-00464-f006:**
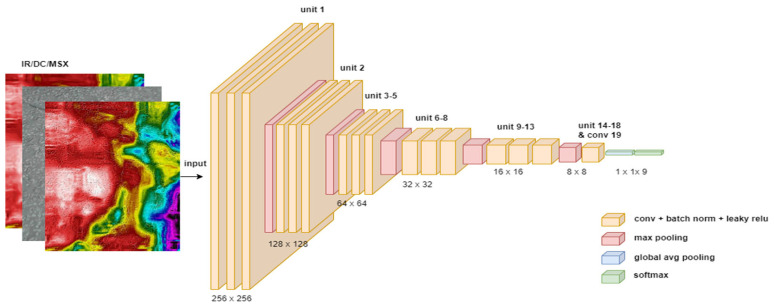
Structure of DarkNet neural network.

**Figure 7 sensors-24-00464-f007:**
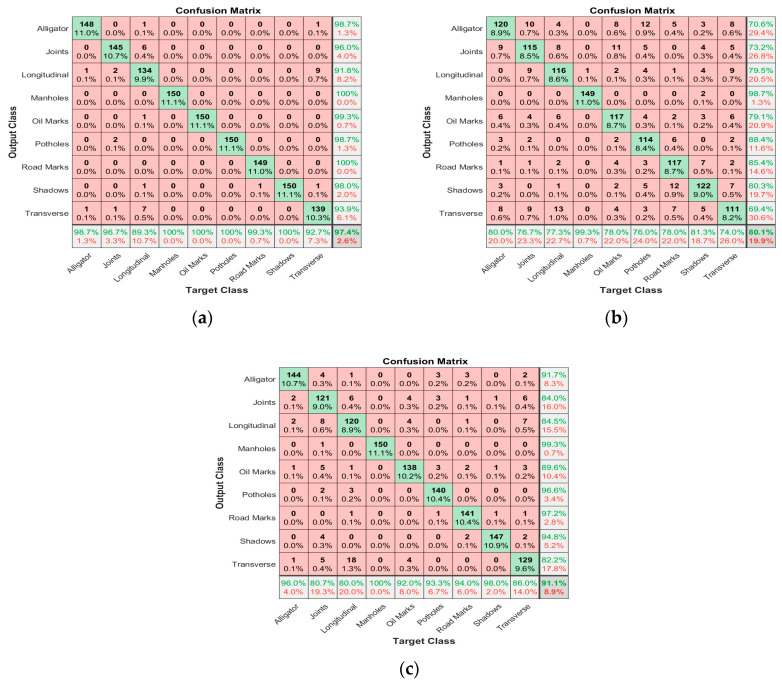
Confusion matrixes of DarkNet. (**a**) Digital image. (**b**) IR image. (**c**) MSX Image.

**Figure 8 sensors-24-00464-f008:**
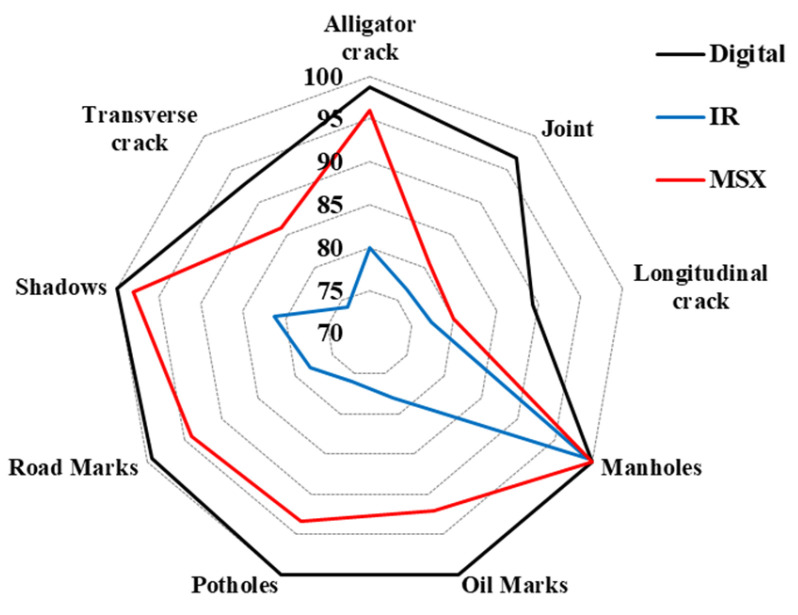
Accuracy comparison of three images.

**Figure 9 sensors-24-00464-f009:**
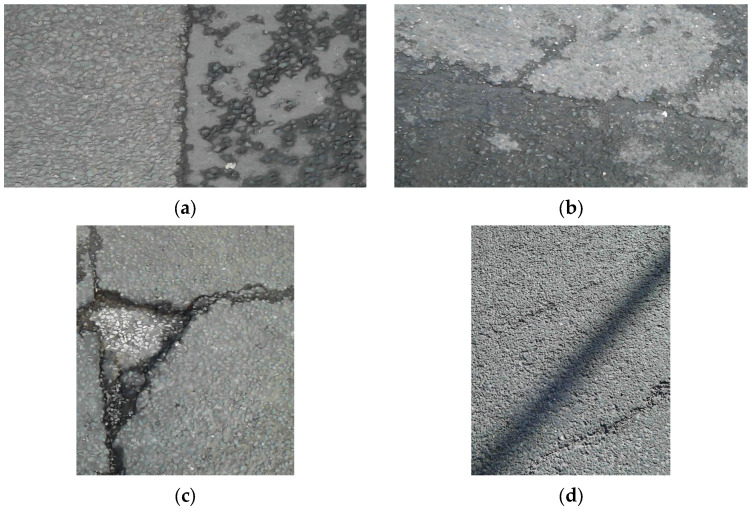
Misclassified digital images. (**a**) Longitudinal crack: misclassified as joint. (**b**) Transverse crack: misclassified as oil mark. (**c**) Pothole: misclassified as longitudinal crack. (**d**) Shadows: misclassified as transverse crack.

**Figure 10 sensors-24-00464-f010:**
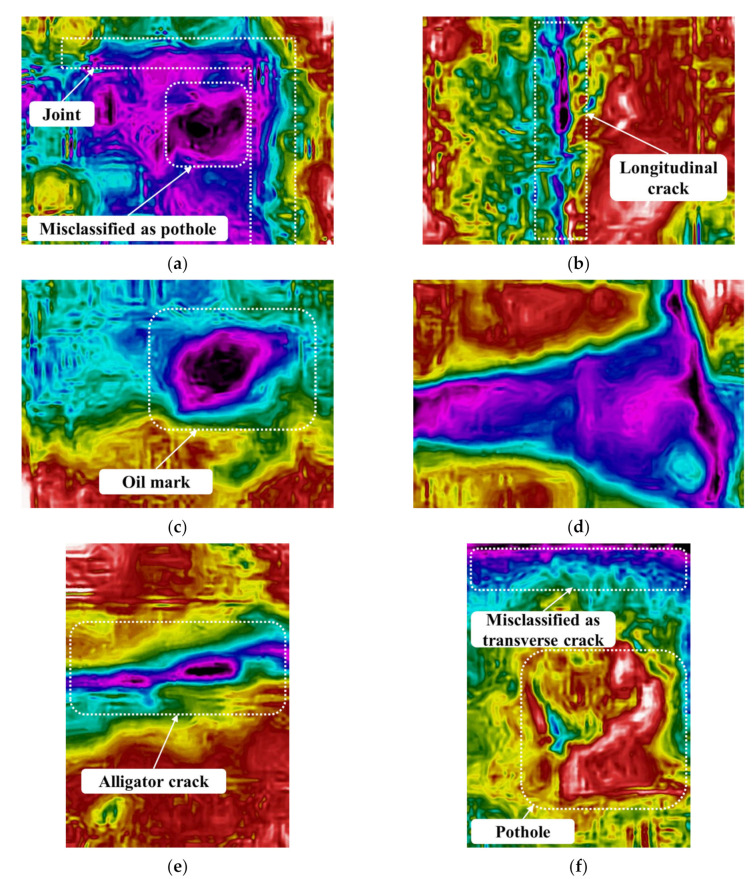
Misclassified IR images. (**a**) Joint: misclassified as pothole. (**b**) Longitudinal crack: misclassified as shadow. (**c**) Oil mark: misclassified as pothole. (**d**) Road marks: misclassified as shadow. (**e**) Alligator crack: misclassified as shadow. (**f**) Potholes: misclassified as transverse crack.

**Figure 11 sensors-24-00464-f011:**
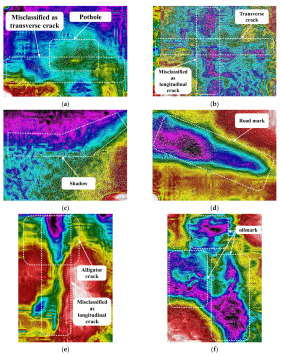
Misclassified MSX images. (**a**) Potholes: misclassified as transverse crack. (**b**) Transverse crack: misclassified as longitudinal crack. (**c**) Shadows: misclassified as transverse crack. (**d**) Road marks: misclassified as transverse crack. (**e**) Alligator: misclassified as longitudinal crack. (**f**) Oil mark: misclassified as pothole.

**Figure 12 sensors-24-00464-f012:**
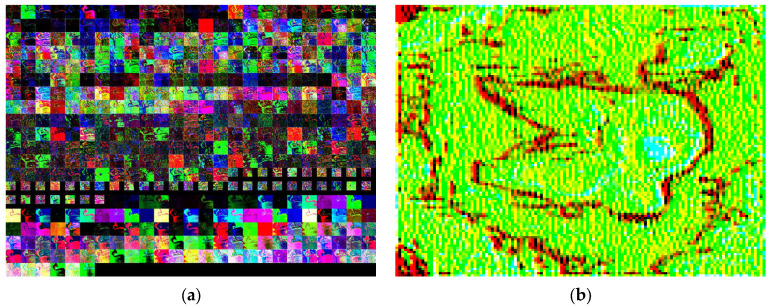
IR images in each layer of DarkNet. (**a**) Image slices of cubes per layer. (**b**) Image output of layer 7.

**Figure 13 sensors-24-00464-f013:**
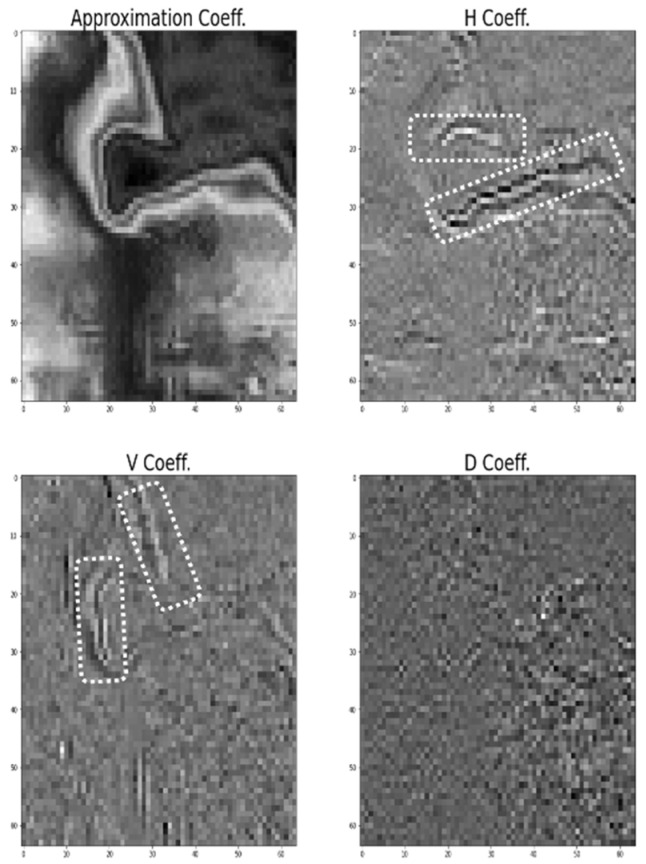
Example of DWT in IR image of joint.

**Figure 14 sensors-24-00464-f014:**
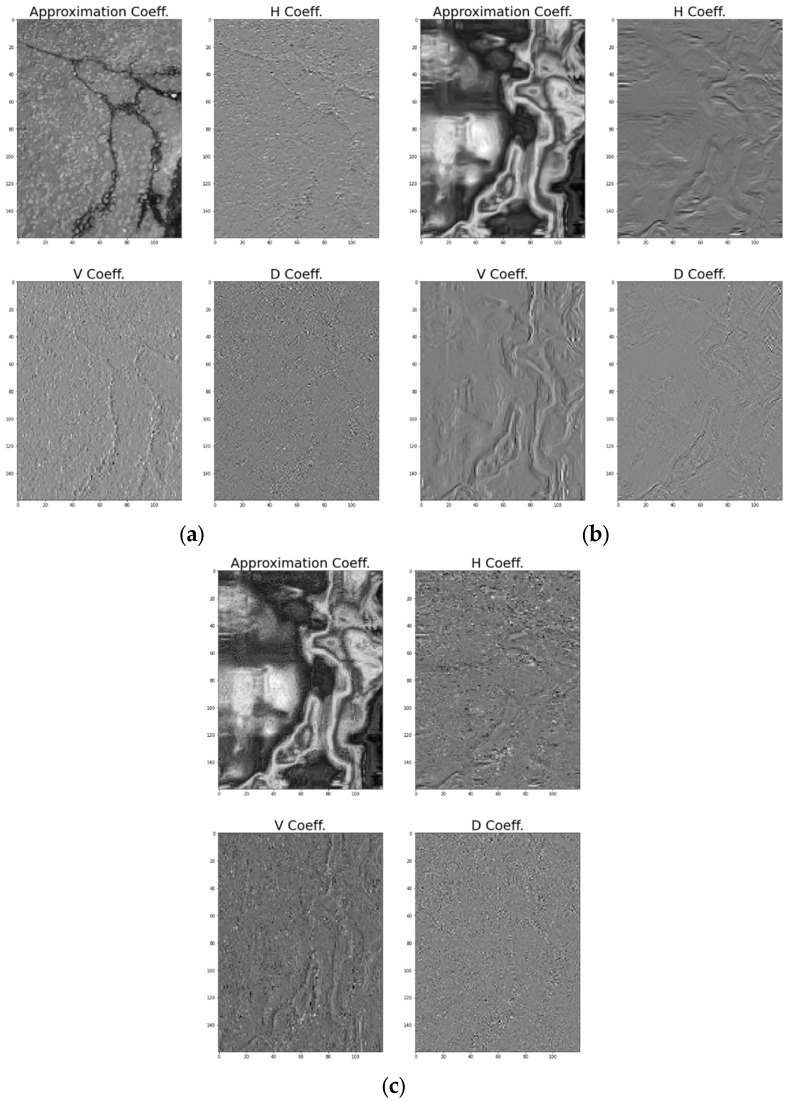
DWT analysis of alligator crack. (**a**) Digital image. (**b**) IR image. (**c**) MSX image.

**Table 1 sensors-24-00464-t001:** Specification of infrared thermal camera.

Parameter	Technical Specification
Dimension (H × W × D)	68 × 34 × 14 mm
Compatibility	Android (USB-C)
Operating temperature	0 °C to 35 °C
Weight	36.5 g
Emissivity Settings	Matte—95%, Semi-Matte—80%, Semi-Glossy—60%, Glossy—30%
Focus	From 15 cm of the object
Frame Rate	8.7 Hz
Horizontal/Vertical Field of View	50° ± 1°/38° ± 1°
Thermal Resolution	80 × 60
Thermal Sensitivity	100 mK
Thermal Sensor	Pixel size 17 µm, 8–14 µm spectral range
Visual Resolution	1440 × 1080
Accuracy	±3 °C or ±5%, typical percent of the difference between ambient and scene temperature. Applicable 60 s after start-up when the unit is within 15–35 °C and the scene is within 5–120 °C.

## Data Availability

The data that support the findings of this study are available from the corresponding author, Hyungjoon Seo, upon reasonable request.

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
