# Peer review of "Automatic Damage Detection of Pavement through DarkNet Analysis of Digital, Infrared, and Multi-Spectral Dynamic Imaging Images"

_sensors, 2024, doi:10.3390/s24020464_

Round 1
Reviewer 1 Report
Comments and Suggestions for Authors
1,The contribution and innovation of the manuscript should be clarified clearly in abstract and introduction.
2,Figures should have high quality.
3,Please compare the advantages/disadvantages of other approaches etc.
4,More future research should be included in conclusion part.
Comments on the Quality of English LanguagePlease improve English. Grammar is Ok but some sentences were made different than a native speaker would make.
Author Response
Thanks for the reviewer's comments and the authors' responses were uploaded.

Reviewer 2 Report
Comments and Suggestions for Authors
In the case of the discrete wavelet transform, the authors did not specify what wavelet was used. The choice of wavelet influences the behavior information contained in the signal. All right the selected wavelet retains its values determined and rejects random.
Author Response

(The authors gave the same response as above.)

Reviewer 3 Report
Comments and Suggestions for Authors
The article titled " Automatic Damage Detection of Pavement through Dark Net 2 Analysis of Digital, IR, and MSX Images" To my observations, the article’s novelty and originality are impressive.
But, I am of the opinion that:
1) Clarify the main objective of the research. Clearly state the purpose of automatically measuring and repairing road pavement damages.
2) Acknowledge the diverse nature of road pavement issues beyond damages (e.g., manholes, road marks) and how the proposed method addresses these.
3) If any code is included in the work provide a repository, such as GitHub for reference.
4) Provide more context on the scale of data collection (13,500 images) and explain why this number was chosen. Consider elaborating on the types of road pavements covered.
5) Highlight the high classification accuracies achieved by DarkNet for digital, IR, and MSX images. Discuss the implications of the differences in accuracy and how they might impact practical implementation.
6) Clarify the relationship between digital, IR, and MSX images. Emphasize the potential complementary role of MSX images in conjunction with digital images, explaining the trade-offs in accuracy.
7) Elaborate on the two-dimensional wavelet transform method for detecting the directionality of cracks. Provide insights into why this method was chosen and how it contributes to the research objectives.
8) Discuss the practical applications of the research findings, especially how the detected crack directionality contributes to the broader field. Outline potential avenues for future research and improvements.
9) Ensure consistency in terminology (e.g., use of "average DarkNet classification accuracies" and "DarkNet classification") for better clarity.
10) Summarize the key findings and emphasize the contribution of the research to the field of road pavement maintenance. Conclude with a clear call-to-action or implication for future work.
11) Also, there is a need for the citations check as found is not uniform.
12) Grammatical errors present in the manuscript, should properly concentrate and revert it.
13) Under introduction point 2 should be improved.
14) Have you done any parameter optimization with the existing classifiers? If so mention it in the paper.
15) Put a comparison graph for the classification results.
16) Strengthen the results and discussion part of the research paper.
17) Finally
Studies done in this style include encouragement for interdisciplinary work. Although it may the subject of the article is a current and an area to be developed further.
Therefore, I do not think that there will be any disadvantages to publishing it.
Author Response

(The authors gave the same response as above.)
